# Isolation and Functional Analysis of *VvWRKY28*, a *Vitis vinifera* WRKY Transcription Factor Gene, with Functions in Tolerance to Cold and Salt Stress in Transgenic *Arabidopsis thaliana*

**DOI:** 10.3390/ijms232113418

**Published:** 2022-11-02

**Authors:** Wei Liu, Xiaoqi Liang, Weijia Cai, Hao Wang, Xu Liu, Longfei Cheng, Penghui Song, Guijie Luo, Deguo Han

**Affiliations:** 1Suqian Institute of Agricultural Sciences, Jiangsu Academy of Agricultural Sciences, Suqian 223800, China; 2National-Local Joint Engineering Research Center for Development and Utilization of Small Fruits in Cold Regions/Key Laboratory of Biology and Genetic Improvement of Horticultural Crops (Northeast Region), Ministry of Agriculture and Rural Affairs/College of Horticulture & Landscape Architecture, Northeast Agricultural University, Harbin 150030, China; 3Institute of Rural Revitalization Science and Technology, Heilongjiang Academy of Agricultural Sciences, Harbin 150028, China

**Keywords:** Beichun, *VvWRKY28*, abiotic stress, transgenic plant

## Abstract

The grape (*Vitis vinifera* L.) not only has a long history of cultivation, but also has rich nutritional value and high economic value. However, grapes often face many threats in the growth process. For example, low temperature and salt stress restrict the growth status, yield, and geographical distribution of grapes. WRKY, as one of the largest transcription factor (TF) families in plants, participates in the response of plants to stress. *VvWRKY28*, a new zinc finger type transcriptional regulator gene, was isolated from Beichun (*V. vinifera* × *V.amurensis*) in this study. From the subcellular localization results, it can be concluded that *VvWRKY28* was localized in the nucleus. The expression of *VvWRKY28* was enriched in leaves (young and mature leaves), and cold and high salt conditions can induce high expression of *VvWRKY28*. After being transferred into *Arabidopsis*, *VvWRKY28* greatly improved the tolerance of *Arabidopsis* to low temperature and high salt and also changed many physiological and biochemical indicators of transgenic *Arabidopsis* to cope with cold and high salt stimulation. The content of malondialdehyde (MDA) was decreased, but for chlorophyll and proline, their content increased, and the activities of superoxide dismutase (SOD), peroxidase (POD), and catalase (CAT) were improved. In addition, under cold stress, binding with cis-acting elements promotes the expression of downstream genes related to cold stress (*RAB18*, *COR15A*, *ERD10*, *PIF4*, *COR47*, and *ICS1*). Moreover, it also plays an active role in regulating the expression of genes related to salt stress (*NCED3*, *SnRK2.4*, *CAT2*, *SOD1*, *SOS2*, and *P5CS1*) under salt stress. Therefore, these results provide evidence that *VvWRKY28* may play a role in the process of plant cold and salt stress tolerance.

## 1. Introduction

In order to adapt and resist low temperature, high temperature, drought, salinization, nutrient deficiency, and other abiotic stresses, plants have formed a set of unique and complex physiological and molecular networks. When encountering abiotic stress, plants will undergo a series of changes at the physiological level, the plant cell membrane and antioxidant system will be destroyed, the photosynthetic function will be affected, and the accumulation of osmotic regulatory substances will change [1]. At the molecular level, the expression of transcription factors (TFs) plays a key role. TFs can combine with cis-acting elements of target sites to regulate the transcription level of target gene [2], that is, plantadapts to the changing external environment through their own complex regulatory mechanisms. WRKY, as one of the mainTF families in plants, has received more and more attention in plant response to adversity since it was first isolated from the sweet potato (*Dioscorea esculenta*). *WRKY* is a zinc finger type transcription regulator unique to plants. It contains a highly conserved WRKYGQK amino acid sequence at the N-terminus and a conserved zinc finger motif (CX4-5CX22-23HXH or CX7CX23HXC) at the C-terminus [3]. According to the characteristics of zinc finger structure and the number of WRKY domains, WRKY TFs are generally divided into three categories: class I is WRKY TFs with C_2_H_2_ type zinc finger structure and 2 WRKY domains; class II WRKY TF is also the C_2_H_2_ type, but only contains one WRKY domain; class III is WRKY TF containing C_2_HC type zinc finger structure and one WRKY domain [4].

A large number of studies have proved thatthe role of *WRKY* is indispensable in regulating plant flowering [5], fruit maturation [6], leaf senescence [7], signal transduction [8], plant immunity [9], and stress response [10]. Overexpression of *PyWRKY75* promoted the growth of transgenic poplar under natural conditions and cadmium stress and protected the poplar from cadmium poisoning [11]. Zhu et al. isolated a novel WRKY TF *Ahwrky75* belonging to IIC WRKY TF family from the ‘salt tolerant’ peanut M34 mutant. The overexpression of this gene was induced after salt stress treatment, which enhanced the salt tolerance of transgenic peanut [12]. In addition, many hormones such as gibberellin (GA3) and abscisic acid (ABA) can also be mediated by WRKYs to trigger hormone signals related to stress, and ethylene can also regulate the expression of *WRKY* [13,14]. Low temperature stress and ABA treatment can induce high-level expression of cucumber *CsWRKY46* and improve the cold tolerance and ABA desensitization of *Arabidopsis* during seed germination [15].

The damage degree of plants under stress can be intuitively reflected by relevant physiological indicators. Malondialdehyde (MDA) is the final decomposition product of membrane lipid peroxidation, and its content can reflect the degree of stress damage to plants. The accumulation of MDA will lead to membrane lipid peroxidation, destroy the integrity of biological membranes, change the permeability of membranes, and affect the normal physiological and biochemical reactions of plants [16,17]. Proline accumulation plays an important role in osmotic regulation in plant cytoplasm. Under normal conditions, the proline content in plants is very low. When subjected to stress, free proline accumulates rapidly. The more its content increases, the stronger the tolerance of plants to adversity [18]. Chlorophyll content is also one of the important indicators for measuring the stress resistance of plants. Under stress conditions, chloroplast structure is destroyed, and chlorophyll content will decrease significantly [19]. In addition, various abiotic stresses can induce the production of reactive oxygen species (ROS) in plant organelles. However, the excessive accumulation of ROS will cause oxidative damage of biological macromolecules and even cell death. Therefore, how to regulate the steady state of ROS is very important [20]. A large number of studies have proved that SOD, CAT, and POD can effectively scavenge reactive oxygen species and protect the structure of cell membranes. The damage degree of plants after stress can be inferred according to their activities [21,22,23].

TFs require multiple gene interactions in regulating plant responses to various stresses. Studies have shown that WRKY TFs regulate the expression of downstream genes by recognizing and combining W-box (TTGACC/T) in cis-acting elements on the promoter region of downstream genes, thus achieving the regulation of plant stress through various physiological activities [24,25]. *A. thaliana* can specifically activate the expression of *SOS2* genes after receiving salt stress signals [26]. The ROS signal is also involved in plant response to stress. Under salt stress, TFs induce the expression of stress response genes and the genes responsible for ROS detoxification. Under salt stress, the overexpression of *RtWRKY23* not only upregulates the expression of *POD* gene to maintain the balance of ROS in transgenic *Arabidopsis* cells, but also promotes the synthesis of proline by regulating the expression of *AtP5CS1*, *AtP5CS2*, and *AtPRODH2* in transgenic *Arabidopsis*, so that plants can maintain normal growth under high salt environment [27]. The interaction between Ca^2+^ and ROS plays an important role in improving the cold tolerance of plants [28]. Overexpression of *RmICE1* in tobacco enhances cold tolerance by regulating ROS scavenging and expression of stress-responsive genes [29]. Genes associated with salt stress are expressed in two ways: the ABA-dependent pathway and the ABA-independent pathway [30]. Sucrose-non-fermentation-1 related protein kinase-2 proteins (SnRK2s) family members play important roles in plant ABA signaling pathways and responses to stress. Studies have shown that *SnRK2.4* affects the growth of *Arabidopsis* taproot salt concentration (when too high) and thus regulates the response of *Arabidopsis* to salt stress [31]. Studies have shown that overexpression of *ClWRKY20* can up-regulate many genes related to low temperature stress, such as *COR27*, *COR413*, *ERD7*, *ERD15*, *ERD10*,and so on [32].

During the growth and development of Beichun, it will inevitably be affected by the natural environment, resulting in failure to achieve the production target [33]. Therefore, it is particularly important to improve the stress resistance of grapes. Based on the important role of WRKY TFs in plant stress response, this study isolated and cloned a novel WRKY TF gene named *VvWRKY28* from the grape Beichun and verified the main functions of *VvWRKY28* under low temperature and salt stress. This finding lays a certain foundation for improving the cold and salt tolerance of grapes by using the regulatory effect of TFs.

## 2. Results

### 2.1. Cloning and Sequence Analysis of VvWRKY28

Appendix A showed the analysis results of ProtParam (http://www.expasy.org/tools/protparam.html, accessed on 1 October 2021). It can be concluded that the *VvWRKY28* gene encoded 319 amino acids, and the complete open reading frame (ORF)was 960 bp. The theoretical pIof the predicted protein was 6.76 and the predicted molecular weight (MW) was 35.00 kDa. The average hydrophilic coefficient was −0.782, which was a hydrophilic protein. The VvWRKY28 protein contained 22 kinds of amino acids, of which Ser, Gly, Pro, Lys, and Asp were more abundant, accounting for 11.6%, 8.8%, 8.5%, 6.6%, and 6.3%, respectively. The underlined part was a WRKY domain and a C_2_H_2_-type zinc finger structure, indicating that it belonged to class II.

### 2.2. Phylogenetic Relationship of VvWRKY28 with Other WRKY Proteins

In order to explore the evolutionary relationship between plant WRKY proteins, WRKY proteins of 14 other species were compared with VvWRKY28 using DNAMAN. These genes were obtained by similarity analysis using blast functions in the NCBI website, including the genes JrWRKY28 (*Juglans regia*, XP_035543967.1), CiWRKY28 (*Carya illinoinensis*, XP_042940078.1), CaWRKY28 (*Coffea arabica*, XP_027088028.1), CsWRKY28 (*Camellia sinensis*, AYA73393.1), HbWRKY28 (*Hevea brasiliensis*, XP_021644344.1), MeWRKY28 (*Manihot esculenta*, XP_021598441.1), ArWRKY28 (*Actinidia rufa*, GFY88621.1), QlWRKY28 *Quercus lobata*, XP_030968701.1), AtWRKY28 (*Arabidopsis thaliana*, NP_193551.1), TcWRKY28 (Theobroma cacao, EOY27787.1), RcWRKY28 (*Ricinus communis*, XP_002519733.1), DzWRKY28 (*Duriozibethinus*, XP_022715624.1), and PtWRKY28 (*Populus trichocarpa*, XP_024436877.1). The comparison results were shown in Figure 1A. The sequence in the red box was the conserved amino acid sequence of the VvWRKY protein, which was unique to the WRKY family. WRKY amino acid sequences of various species had high homology only in their conserved domains. According to the homology of the phylogenetic tree (Figure 1B), VvWRKY28 had the highest homology with JrWRKY28 and Ci WRKY28 and has the closest evolutionary relationship.

### 2.3. Subcellular Localization of VvWRKY28Protein

Figure 2 showed the subcellular localization results of the VvWRKY28 protein. It can be seen that GFP, which was used as a control, was distributed in the whole cytoplasm (Figure 2B), while VvWRKY28 GFP fusion protein only appeared in the nucleus (Figure 2E). Meanwhile, the position of the nucleus was also confirmed after 4′,6-diamino-2-phenylindole (DAPI) staining (Figure 2F). These results indicated that the VvWRKY28 protein was localized in the nucleus.

### 2.4. Expression Specificity Analysis of VvWRKY28

Under control conditions, the expression levels of the *VvWRKY28* gene in different tissues can be seen from Figure 3. The results of qRT-PCR showed that *VvWRKY28* was expressed differently in various tissues, and the expression level from high to low was in new leaves, roots, mature leaves, and stems. Among them, the expression levels in new leaves were 2.08-fold, 2.48-fold, and 2.74-fold higher than those in roots, stems, and mature leaves, respectively. This suggested that the expression of *VvWRKY28* was tissue-specific.

The *VvWRKY28* gene was subjected to stress treatment under the conditions of cold, high salt, water shortage, and high temperature. The expression of the *VvWRKY28* gene after treatment for 1, 3, 6, 12, 24, and 48 h was shown in Figure 3B,C. Under cold conditions, in new leaves, the expression of *VvWRKY28* began to be up-regulated after 1 h of stress, and reached the maximum at 3 h, about 9-fold that of the control group (CK). After that, the expression began to decline, and in roots, it reached the maximum at 12 h. Under salt stress, the time point when the expression of *VvWRKY28* reached the highest level in new leaves and roots was 12 h and 6 h, respectively, and the expression amount was 10-fold and 6-fold that of the untreated group. Under drought and heat stress, the expression of *VvWRKY28* in new leaves reached the maximum at 24 h and 12 h, respectively, whilein the root, it reached the peak at 12 h and 24 h. However, when the ambient temperature was low or the salt concentration was high, the maximum expression level of *VvWRKY28* in new leaves and roots was higher than that in drought or high temperature environments.

### 2.5. Analysis of the Ability of VvWRKY28 Transgenic A. thaliana to Cold Stress

The results of qRT-PCR showed that *VvWRKY28* was more sensitive to the stimulation of cold and salt, so the transgenic *A. thaliana* was obtained through the control of the CaMV 35S promoter, and it was placed in cold and high salt environment to explore the role of *VvWRKY28* under these two stresses. Using wild-type plants (WT) as controls, qRT-PCR was performed on T_2_ generation transgenic line *A. thaliana*. The result showed that eight lines were successfully transformed, and the expression level of *VvWRKY28* was relatively high in Strain 2, Strain 6, and Strain 7 (S2, S6 and S7) (Figure 4A).

Figure 4B showed the phenotype of WT *A. thaliana* and transgenic *A. thaliana* (S2, S6, S7) after low-temperature treatment. It can be seen that there was no obvious difference in the growth trend of all lines without cold treatment. However, after growing at low temperature (−4 °C) for 12 h, the growth trend of WT became weak, and the leaf wilting was obvious, while the leaf damage of transgenic lines was not obvious.

The survival rate of all plants (WT, S2, S6, S7) was calculated after being put back to normal conditions for five days (Figure 4C). Under the control condition, the survival rates of WT and transgenic *A. thaliana* were basically at the same level—97.1%, 96.5%, 97.4%, and 98.3%, respectively. After growing at −4 °C for 12 h, the survival rate of WT decreased significantly (only 18.57% of the plants survived), while the survival rates of transgenic lines were 77.8%, 68.8%, and 82.3%, respectively. There was a significant difference between the WT and transgenic *A. thaliana*. This result indicated that transgenic plants were more tolerant to cold damage.

The content of some physiological indicators and the activity of related enzymes were also measured. The purpose of this was to further clarify the role of *VvWRKY28* under low temperature stress. As shown in Figure 5, the contents of MDA, chlorophyll, and proline, as well as the activities of CAT, POD, and SOD of all plants, were almost equal under control conditions in all plants. After cold stress treatment, the MDA content of WT *A. thaliana* was significantly higher than that of transgenic lines, while the content and activity of other indicators were higher in transgenic *A. thaliana*. The above results indicatedthat the transgenic plants had a stronger antioxidant capacity, a lower degree of plasma membrane peroxidation, a stronger ability to scavenge ROS, and can survive better at low temperature, while WT plants had no such ability.

### 2.6. Expression Analysis of Cold-Resistant Downstream Genes in VvWRKY28-OE A. thaliana

WRKY TFs can specifically bind to the W-box of the promoter of the target gene, thereby playing an active or inhibitory regulatory role on the expression of downstream genes. WRKY TFs play an important regulatory role in the response of plants to low temperature stress. Therefore, the expression changes of six key genes located downstream of WRKY TF were analyzed (Figure 6), including *AtRAB18* (*Arabidopsis thaliana*, U75603.1), *AtCOR15A* (*Arabidopsis thaliana*, NM_001336986.1), *AtERD10* (*Arabidopsis thaliana*, BG732218.1), *AtPIF4* (*Arabidopsis thaliana*, NM_001337007.1), *AtCOR47*(*Arabidopsis thaliana*, NM_101894.4), and *AtICS1* (*Arabidopsis thaliana*, AT1G74710). *A. thaliana* was grown at −4 °C for 12 h, and it was found that these six genes had higher expression levels in the *VvWRKY28* transgenic line. The expression of *AtRAB18*, *AtCOR15A*, *AtERD10*, *AtPIF4*, *AtCOR47*,and *AtICS1* was actively up-regulated, thereby further enhancing the adaptability of plants to cold.

### 2.7. Analysis of the Ability of VvWRKY28 Transgenic A. thaliana to Resist Salt Stress

Only 200mm NaCl solution was used to irrigate WT and transgenic *A. thaliana* transplanted into nutrient soil every day. After a week, the growth of all plants was observed. It was found that the phenotypes of WT and transgenic plants were healthy under control conditions. After seven days of salt stress treatment, the leaves of all plants appeared yellowing, but the withering of transgenic plants was not obvious, and the yellowing of wild-type plants was serious (Figure 7A).

Figure 7B showed the survival rates of all *A. thaliana* strains grown in a high salt environment. When the growth conditions were normal, the survival rates of WT and transgenic lines (S2, S6, and S7) were 97.3%, 96.7%, 96.4%, and 97.2%, respectively. The survival rates of WT and transgenic plants (S2, S6 and S7) were significantly different after surviving in a high salt environment for one week, and the survival rates decreased to 22.5%, 85.4%, 84.7%, and 84.6%, respectively. Compared with WT plants, the survival rate of transgenic plants under salt stress was significantly higher.

The changes of various physiological indexes of *A. thaliana* after salt stress treatment were shown in Figure 8. Under control conditions, the contents of chlorophyll, MDA, and proline, and the activities of SOD, POD, and CAT of all plants, were almost at the same level. However, under high salt conditions, the overexpression of *VvWRKY28* led to a significant increase in the activities of SOD, POD, and CAT in transgenic *A. thaliana*, a significant increase in the content of proline, and a slight decrease in the content of chlorophyll, but it was still much higher than the that of WT plants. Moreover, it was obvious that the MDA content of transgenic lines was lower than that of WT lines. Therefore, it can be concluded that the high expression level of *VvWRKY28* had a significant effect on improving the salt tolerance of transgenic *A. thaliana*.

### 2.8. Expression Analysis of Salt-Resistant Downstream Genes in VvWRKY28-OE A. thaliana

The signal transmission of salt stress is the core issue of plant salt stress response. TFs play a key role in the signal transmission pathway. No matter which pathway, the response of TFs to salt stress can be through ABA-dependent and ABA-independent pathways. Therefore, in order to investigate the effect of WRKY TFs on the expression of downstream salt stress-related genes, the expression levels of six salt stress-related genes were examined. These six genes were *AtNCED3* (*A. thaliana*, NM_112304.3), *AtSnRK2.4* (*A. thaliana*, AT1G10940), *AtSOS2* (*A. thaliana*, NM_001344103.1), *AtP5CS1* (*A. thaliana*, AB022784.2), *AtCAT2* (*A. thaliana*, AT1G58030),and *AtSOD1*(*A. thaliana*, AT1G08830). The change of expression amount was shown in Figure 9. There was no obvious difference in the expression amount of these genes without salt stress treatment. After growing in a high salt environment (200 mm NaCl) for seven days, *AtNCED3*, *AtSOS2*, *AtSnRK2.4*, *AtP5CS1*, *AtCAT2*, and *AtSOD1* in *VvWRKY28*-OE linesincreased more. These changes can explain that *VvWRKY28* can improve the salt tolerance of plants through a variety of signal transduction pathways. The SOS signal transduction pathway is an important pathway for plants to respond to salt stress, which can regulate ion homeostasis at the cellular level to improve plant salt tolerance. *SOS2* is a key gene in this pathway.

## 3. Discussion

In recent years, more and more TFs have been identified and studied, proving that they play a key role in plant abiotic stress. Whether at the cellular or individual level, transcription factor regulation is a very important way to control gene expression [26,34]. As one of the largest TF families in plants, WRKY TFs play an indispensable role in regulating plant responses to stress. Numerous studies have shown that WRKY TFs play an important role in the regulatory mechanism and can significantly enhance the resilience of plants to cope with complex environments [26]. Understanding the important regulatory role of *WRKY* genes in plant defense against adverse environments is critical for plant breeding and crop improvement.

In this study, the WRKY gene fragment was obtained by designing specific primers using homologous cloning technology, and the cloned target gene was analyzed by bio-informatics to explore the basic information of the gene and predict its biological function. After sequence analysis with DNAMAN, it was found that the amino acid sequence of the TF contains a highly conservative heptapeptide sequence WRKYGQK, and only contained a WRKY domain and a C_2_H_2_ zinc finger structure, indicating that it was a class II member of the WRKY family. Although the length of the VvWRKY28 protein was not completely the same as that of other homologous WRKY proteins, they differed greatly in non-conservative sequences, and they contained highly similar conservative sequences, which indicated that the *WRKY* gene family was highly conservative in the evolution process. The proteins with the highest homology and closest evolutionary relationship with VvWRKY28 were JrWRKY28 and CiWRKY28 were considered, which meant they had the same function. Class II WRKY TFs are generally involved in regulating plant growth and evolution and also play a role in plant resistance to their stress environment. It has been found that *CiWRKY* can respond to at least one abiotic stress (drought, cold, salt, high pH, or ABA) [35]. We also found that most of the homologous genes of *VvWRKY28* are related to abiotic stresses such as low temperature and high salt. Therefore, we speculate that the target genes studied are also likely to participate in the process of plants responding to stress.

The WRKY TF geneisactively involved in the regulation of plant growth and metabolism and the transduction of various stress signals to regulate plant responses to stress [24,36]. At present, the research on the regulation of WRKY TFs on abiotic stress in plants is still in progress. In a previous study, we found that we found that overexpression of the *MbWRKY1* gene isolated from *M. baccata* led to enhanced tolerance of transgenic tobacco to drought stress [37]. He et al. found that *TaWRKY33* from the wheat subgroup IIa played an active regulatory role in ABA and drought-responsive signaling networks [38]. Multiple *CmWRKY* genes were rapidly up-regulated under drought and NaCl treatment, responding significantly to these stresses [39]. Therefore, the target gene *VvWRKY28* in this study is very likely to play a similar role in the network of plant stress tolerance mechanisms.

As can be seen from Figure 3A, *VvWRKY28* can be induced in roots, stems, new leaves, and mature leaves, but the expression levels are different, and higher expression levels are found in new leaves and roots. This indicated that the expression of the gene was tissue-specific, which may be because *VvWRKY28* mainly senses stress signals in active organs.

In new leaves and roots, the expression of the *VvWRKY28* gene after low temperature, high salt, water shortage, and high temperature stress was shown in Figure 3B,C. When the new leaves were grown under these four growth conditions, the *VvWRKY28* gene expression level reached the peak time of 3 h, 12 h, 24 h, and 12 h, respectively. Under the same conditions, *VvWRKY28* gene expression in roots reached the highest level at 12, 6, 12, and 24 h, respectively. This situation may be because new leaves were more sensitive to low temperature and high temperature stress than roots, indicating that new leaves were more susceptible to temperature, while roots respond faster to high salt and drought stress. Moreover, the maximum expression level of *VvWRK28* in new leaves and roots under cold and salt stress was higher than that under other stresses. These phenomena indicated that *VvWRK28* gene expression could be induced by low temperature, high salt, drought, and high temperature stress, but *VvWRKY28* might play a more obvious role in regulating plant response to low temperature and salt stress.

Abiotic stress will not only change the appearance of plants and affect their survival rate, but it will also interfere with the normal metabolic pathways of plants, making various physiological indicators of plants not at normal levels during their growth and development [40,41]. Free radicals are inevitably produced in aerobic biological cells. The excessive accumulation of free radicals will cause harmful effects on plants, accelerate cell aging and damage, and lead to diseases [42]. The protective enzyme system is an important way to scavenge reactive oxygen species. This system mainly includes SOD, POD, and CAT. Their activities can be used as very key indicators in the study of plant resistance [43]. The growth environment of plants will affect the content of chlorophyll, which can reflect the growth status of plants in adversity. Therefore, the content of chlorophyll is of great significance to study the response of plants to adversity [44,45,46]. The accumulation of osmotic substances helps to improve the tolerance of plants to stress to a great extent. As a typical osmotic substance, proline plays an indispensable role in regulating osmotic balance and protecting cell structure and participates in the response process of plants to stress [47,48]. The accumulation of MDA will damage the plant cell membrane and cells to a certain extent. There is a close relationship between its content and the damage degree of plants. The more MDA that is accumulated, the more serious the damage to cell membranes will be [49,50]. The overexpression of *VvWRKY28* in transgenic *A. thaliana* improvedthe survival rate of these plants under the survival pressure of cold and high salt (Figure 5C and Figure 7B), and also significantly increased the activities of SOD, CAT, POD, and the content of proline. Although the content of chlorophyll decreased slightly, it was significantly higher than that of WT. The content of MDA decreased greatly. These results all indicated that, when plants were subjected to low temperature and salt stress, *VvWRKY28* can increase the accumulation level of enzymatic antioxidants (SOD, CAT, and POD) and proline, prevent the excessive accumulation of reactive oxygen species, and maintain cell osmotic balance, thereby improving plant cold tolerance and salt tolerance. In addition, lower MDA content can reduce the amount of oxidative damage to plant cells, so transgenic lines will be more efficient than WT lines.

As a plant-specific TF, the WRKY family is widely involved in plant response to stress. At the same time, the expression of genes can be regulated by binding to cis-acting elements in the promoter regions of downstream genes to improve abiotic stress tolerance [51]. After low temperature treatment, it was found that the expression levels of the cold stress-related genes *AtRAB18*, *AtCOR15A*, *AtERD10*, *AtPIF4*, *AtCOR47*, and *AtICS1* located downstream of *VvWRKY28* were higher than those without cold stress, especially in *VvWRKY28*-OE plants (this is more obvious in Figure 6). The SOS signal transduction pathway is an important pathway for plants to respond to salt stress, which can regulate ion homeostasis at the cellular level to improve plant salt tolerance. *SOS2* is a key gene in this pathway [26]. ROS is a key signaling molecule for cells to rapidly respond to different stimuli. Many studies have demonstrated that WRKY TFs can regulate the homeostasis of ROS by regulating the expression of antioxidant enzymes (CAT, SOD, POD) genes and improve the salt tolerance of plants [52,53]. TFs regulate salt stress mainly through ABA-dependent and ABA-independent pathways. Compared with WT *Arabidopsis*, under salt stress, the expression levels of *SOS2* and *P5CS1* genes and the correlation in ABA signal transduction pathway genes *NCED3* and *SnRK2.4*, as well as the antioxidant enzyme genes *CAT2* and *SOD1*, were more significantly up-regulated in *VvWRKY28*-OE *A. thaliana*. Thus, the adaptability of plants to salt stress was improved. The expression levels of these genes were generally consistent with previous studies [54,55]. These results suggested that plants can not only improve their tolerance to low temperature by combining with cis-acting elements of downstream target genes, but also regulate the expression of downstream genes related to salt stress through ABA-dependent and ABA-independent pathways, as well as ROS and SOS pathways, so as to make plants more adapted to high salt environment. The high-level expression of these related downstream genes proved that *VvWRKY28* plays an important regulatory role in the complex mechanism of plant response to salt and cold stress, which laid a foundation for further studies on the function of WRKY TFs. Besides, further studies are needed to understand the molecular mechanism of *VvWRKY28* in regulating plant stress responses.

Therefore, according to previous reports and the results of this experiment, a possible model was established to show the response mechanism of *VvWRKY28* gene to low temperature and salt stress (Figure 10). The expression level of *VvWRKY28* was significantly increased after cold stress, and it was bound to the W-box on the promoters of downstream target genes and regulated their expression, resulting in a significant increase in the expression levels of *RAB18*, *COR15A*, *ERD10*, *PIF4*, *COR47*, and *ICS1*. In addition, salt stress can also induce the high-level expression of *VvWRKY28*, activate the ABA signaling pathway and the salt stress response process, and promote the expression of *NCED3* and *SnRK2.4* in the ABA-dependent pathway. *CAT2*, *SOD1*, and *SOS2* were rapidly expressed to regulate the content of ROS and ion balance in cells, and the expression level of ABA-independent *P5CS1* gene was significantly increased.

## 4. Materials and Methods

### 4.1. Plant Material and Growth Conditions

The tissue culture seedlings of Beichun(it is a hybrid of *Muscat Hamburg* and *Vitis amurensis* Rupr by the Beijing Botanical Garden, Institute of Botany, Chinese Academy of Sciences) were placed in a MS growth medium containing agar for rapid propagation. The concentration of indole-3-butyric acid (IBA) and 6-benzylaminopurine (6-BAP) added to the medium was 0.6 mg/L. After 35 days of culture, tissue culture seedlings with good growthwere transferred to a MS rooting medium for continuous culture to promote their rooting. At this time, the concentration of IBA added to the medium was 1.2 mg/L [23]. After growing roots, the tissue cultured seedlings were transplanted to Hogland nutrient solution for growth, and the hydroponic solution was changed every 3 days. The environment of the tissue culture room remained stable (temperature of about 25 °C, relative humidity of 80–85%). When the tissue culture seedlings grow to have 7–10 complete leaves (fully expanded leaves) and relatively strong roots, the 50 seedlings with good morphology and basically the same growth trend were divided into 5 groups, which were respectively placed under low temperature (incubator temperature of 4 °C), high salt (200 mM NaCl high saline culture conditions), water shortage (Hogland nutrient solution containing PEG6000 with a concentration of 20%), and high temperature (incubator temperature of 37 °C). The other group, as the CK, was grown in a tissue culture chamber at 25 °C and hydroponically cultured with normal Hoagland solution. According to our previous study, the new leaves, roots, mature leaves, and stems of all seedlings were sampled at 0, 1, 3, 6, 12, 24, and 48 h after treatment [15]. After sealing, they were immediately placed in liquid nitrogen and stored at −80 °C to prepare for RNA extraction.

### 4.2. Cloning and qRT-PCR Expression Analysis of VvWRKY28

The EasyPure Plant RNA Kit (TransGen Biotech, Beijing, China) was used to extract total RNA. The extracted parts were roots, stems, young leaves, and mature leaves. The first strand of cDNA was synthesized by Trans Script^®^ First-Strand cDNA Synthesis Super Mix (transgen, Beijing, China). With the CDs region of *VvWRKY28* as the reference sequence, two pairs of specific primers were designed with Primer 5.0 software, named *VvWRKY28*-F and *VvWRKY28*-R, respectively. The specific sequences were shown in Appendix A. The complete sequence of the gene was obtained by PCR, and the amplification template was the first strand cDNA of Beichun. After gel purification, the obtained DNA fragment was cloned into pEASY-T1 Cloning Kit (TransGen Biotech, Beijing, China) and sequenced (BGI, Beijing, China).

With reference to the method of Han et al. [56], qRT-PCR was performed on *VvWRKY28* to analyze its expression under abiotic stress. The housekeeping gene *VvActin* (AF369524) was used as an internal control, and the expression of this gene can remain stable under almost all conditions. 25 μL of qPCR amplification mixture contained 12.5 μL 2xMix, 9 μL ddH_2_O, 1.5 μL cDNA, 1 μL *VvWRKY28*-F, and 1 μL *VvWRKY28*-R. The PCR reaction procedure was: pre denaturation at 95 °C for 5min, denaturation at 95 °C for 45 s, annealing at 56 °C for 1 min, and extension at 72 °C for 1 min. After 35 cycles, it continued to extend at 72 °C for 5 min [57]. The relative expression of the gene was obtained bythe 2^−ΔΔCT^ method [58].

### 4.3. Bioinformatics Analysis of the VvWRKY28 Gene

The primary structure and various physicochemical properties of the protein were predicted by ExPASy (https://web.expasy.org/protparam/). After this, the carried out sequence similarity analysis was analyzed by the Blast program in the NCBI database (http://www.ncbi.nlm.nih.gov./, accessed on 13 June 2022). Several WRKY sequences with high homology to VvWRKY28 from different species were selected and compared using DNAMAN 8.0 software. These amino acid sequences included JrWRKY28 (*Juglans regia*, XP_035543967.1), CiWRKY28 (*Carya illinoinensis*, XP_042940078.1), CaWRKY28 (*Coffea arabica*, XP_027088028.1), CsWRKY28 (*Camellia sinensis*, AYA73393.1), HbWRKY28 (*Hevea brasiliensis*, XP_021644344.1), MeWRKY28 (*Manihot esculenta*, XP_021598441.1), ArWRKY28 (*Actinidia rufa*, GFY88621.1), TcWRKY28 (*Theobroma cacao*, EOY27787.1), RcWRKY28 (*Ricinus communis*, XP_002519733.1), and DzWRKY28 (*Duriozibethinus*, XP_022715624.1), PtWRKY28 (*Populus trichocarpa*, XP_024436877.1). MEGA 7 was used to construct a phylogenetic tree.

### 4.4. Subcellular Localization Analysis of the VvWRKY28 Protein

The *VvWRKY28* ORF was cloned between the *SacI* and *BamHI* sites of the pSAT6-GFP-N1 vector, and the modified red-shifted green fluorescent protein (GFP) was present between these two sites. The constructs were injected into onion epidermal cells, the nuclear markers in the nuclear assay were stained with DAPI, and the transient expression of *VvWRKY28*–GFP fusion protein was observed under confocal microscopy (LSM 510 Meta, Zeiss, Jena, Germany).

### 4.5. Obtaining Transgenic A. thaliana

The 5′ and 3′ ends of *VvWRKY28* cDNA were added to the restriction sites of *SacI* and *BamHI* by PCR to construct an expression vector for the transformation of *A. thaliana*. The primers used were shown in Appendix A (q-PCR-F, q-PCR-R). The PCR product and PCAM3011 were linked together by GUS gene replacement after being digested by *SacI* and *BamHI*. Transformation of the Agrobacterium, mediated by GV3101, into the Columbia ecotype *A. thaliana* was completed by the inflorescence-mediated method. Transgenic *A. thaliana* was selected for the MS medium (the concentration of kanamycin added was 50 mg/L). After qRT-PCR analysis, the transgenic line was finally determined, and the CK was WT. Further analysis was carried out with T_3_ generation plants as experimental materials.

### 4.6. Determination of Relevant Physiological Indexes

The seeds of WT and transgenic lines (S2, S6, and S7) were sown on the culture medium and then transferred to the nutrient soil containing vermiculite two weeks later (vermiculite: nutrient soil ratio was 1:2). After 14 days of growth, all *A. thaliana* was divided into two groups, one group was treated at the low temperature of −4 °C for 12 h, and the other group was treated at high salt conditions irrigated with 200 mM NaCl solution for 7 days. Then, the survival rate was calculated with 15 nutrient pots [59].

CK and Arabidopsis thaliana, after stress treatment, were collected for the measurement of physiological indicators. The measurement of chlorophyll content referred to the method of Lu et al. [60]. Proline content was measured according to the method of Ou et al. [61]. MDA content was analyzed at 530 nm by the TBA method [62]. CAT activity was measured according to the method described by Zhang et al. [63]. SOD activity was determined with nitroblue tetrazolium (NBT) [64]. The activity of POD was measured according to the method described by Wang et al. [65].

### 4.7. Analysis of the Downstream Gene Expression of VvWRKY28

The RNA of WT and transgenic *Arabidopsis*, without stress treatment and after cold and salt stress treatment, was extracted. The RNA was reverse transcribed into the first strand cDNA as an amplification template. The internal reference was *VvActin*. According to the reaction system in Section 2.2, the expression levels of several downstream target genes (*RAB18*, *COR15A*, *ERD10*, *PIF4*, *COR47*, *ICS1*, *NCED3*, *SnRK2.4*, *CAT2*, *SOD1*, *SOS2*, and *P5CS1*) of WRKY TFs were detected by QRT PCR. The specific primers used were listed in Appendix A.

### 4.8. Statistical Analysis

The one-way analysis of variance was completed by SPSS21.0 software (IBM, Chicago, IL, USA). All data were obtained after 3 repeated tests, and then thestandard deviation (SD) was measured separately. Statistical differences were referred to as significant (* *p* ≤ 0.05, ** *p* ≤ 0.01).

## 5. Conclusions

In this study, a WRKY-TF named *VvWRKY28* was isolated from Beichun. The subcellular results showed that the VvWRKY28 protein was localized in the nucleus. From the phylogenetic tree results, it can be concluded that the genetic relationship between the VvWRKY28 protein and the VvWRKY71 protein was the closest. Low temperature, high temperature, high salt, and drought stress can induce the high-level expression of *VvWRKY28*, but *VvWRKY28* was more sensitive to low temperature and high salt signals, and its expression in roots and new leaves was higher. After it was introduced into *Arabidopsis*, it caused changes in many physiological and biochemical indicators of *Arabidopsis* under low temperature and high salt environment. For example, the content of MDA was significantly lower than that of WT *Arabidopsis*, but the content of proline and chlorophyll and the activities of CAT, SOD, and POD were higher in transgenic *A. thaliana*. In addition, the overexpression of *VvWRKY28* can also play a certain regulatory role in the expression of downstream target genes. Under low temperature stress, binding with cis-acting elements promotes the expression of downstream genes related to cold stress, such as *RAB18*, *COR15A*, *ERD10*, *PIF4*, *COR47*, and *ICS1*. Moreover, it can also positively regulate the expression of genes related to salt stress such as *NCED3*, *SnRK2.4*, *CAT2*, *SOD1*, *SOS2*,and *P5CS1* under salt stress. The results of this study comprehensively showed that the overexpression of *VvWRKY28* played a key role in improving the cold tolerance and salt tolerance of plants.

## Figures and Tables

**Figure 1 ijms-23-13418-f001:**
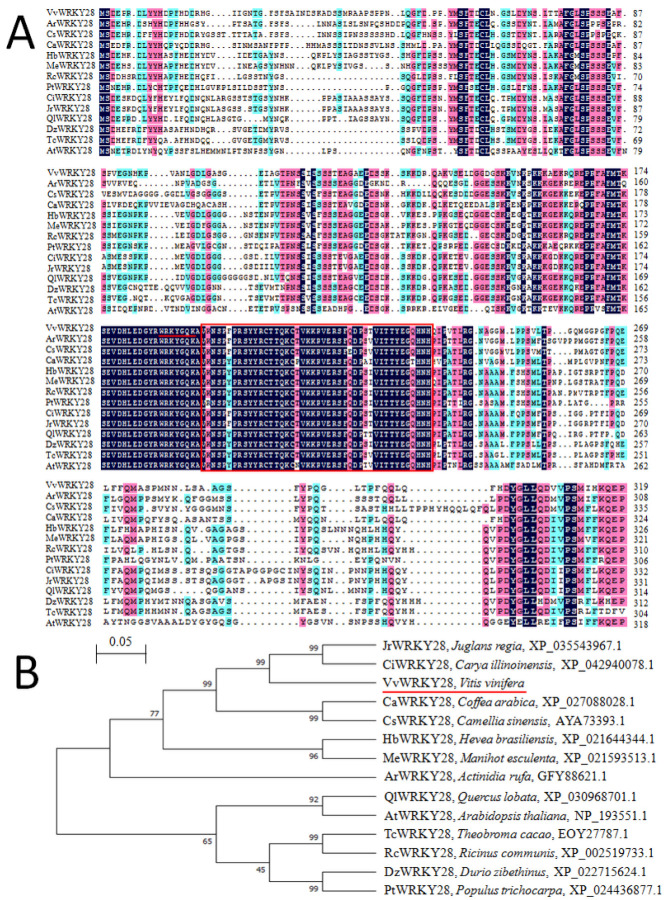
Comparison and phylogenetic relationship of WRKY-TF with proteins from other species. (**A**) Alignment results of VvWRKY28 with WRKY-TF proteins of other species. The conserved sequences were shown in red boxes, and the target proteins were underlined with red lines. (**B**) The phylogenetic tree was constructed by the MEGA-7 adjacency method to analyze the relationship between VvWRKY28 and WRKY-TF proteins in different plants. These include JrWRKY28 (*Juglans regia*, XP_035543967.1), CiWRKY28 (*Carya illinoinensis*, XP_042940078.1), CaWRKY28 (*Coffea arabica*, XP_027088028.1), CsWRKY28 (*Camellia sinensis*, AYA73393.1), HbWRKY28 (*Hevea brasiliensis*, XP_021644344.1), MeWRKY28 (*Manihot esculenta*, XP_021598441.1), ArWRKY28 (*Actinidia rufa*, GFY88621.1), QlWRKY28 *Quercus lobata*, XP_030968701.1), AtWRKY28 (*Arabidopsis thaliana*, NP_193551.1),TcWRKY28 (Theobroma cacao, EOY27787.1), RcWRKY28 (*Ricinus communis*, XP_002519733.1), DzWRKY28 (*Duriozibethinus*, XP_022715624.1), and PtWRKY28 (*Populus trichocarpa*, XP_024436877.1).

**Figure 2 ijms-23-13418-f002:**
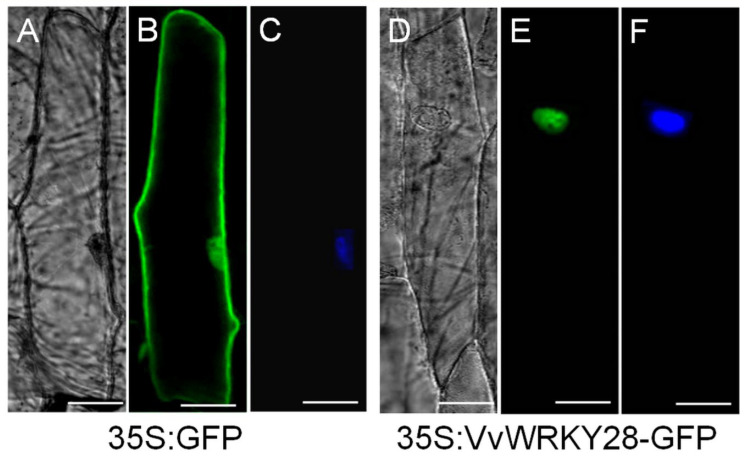
Subcellular localization of the VvWRKY28 protein: the transient expression of green fluorescent protein 35S: GFP and 35S: VvWRKY28-GFP fusion protein in onion epidermal cells was observed with a fluorescence microscope. (**A**,**D**) was the bright field effect, (**B**,**E**) was the dark field effect, and (**C**,**F**) was the result after DAPI staining. The scale corresponds to 5 µM.

**Figure 3 ijms-23-13418-f003:**
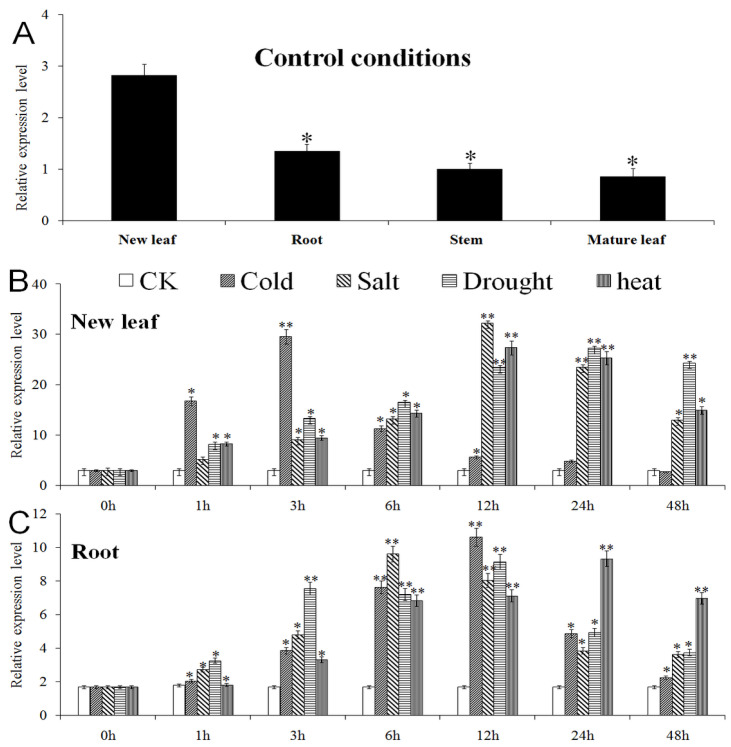
qRT-PCR results of the *VvWRKY28* gene. (**A**) Under normal conditions, *VvWRKY28* was expressed in new leaves, roots, stems, and mature leaves of grapes. The expression level in the new leaves was the control. (**B**,**C**) expression levels of *VvWRKY28* in new leaves and roots at multiple points in time (0, 1, 3, 6, 12, 24, and 48 h) under normal conditions, low temperature, salt, drought, and heat stress. The data were the average of 3 repeated trials. The asterisk above the column indicated that there were significant and extremely significant differences between the treatment group and the control group (control) (* *p* ≤ 0.05; ** *p* ≤ 0.01).

**Figure 4 ijms-23-13418-f004:**
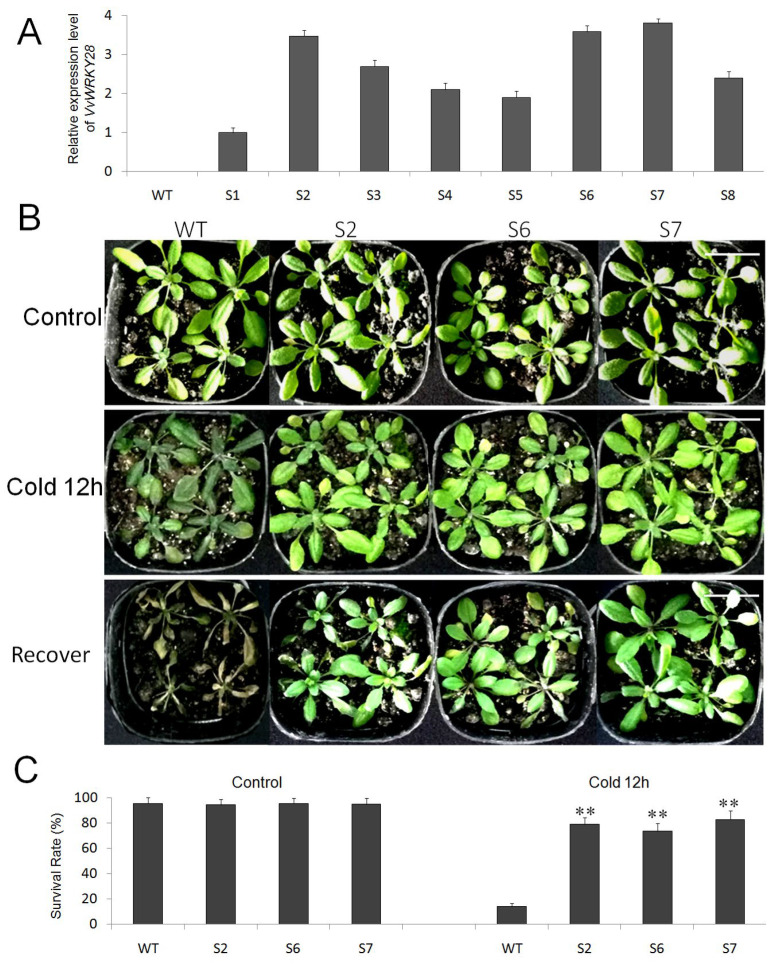
The overexpression of *VvWRKY28* improved the cold tolerance of *Arabidopsis*. (**A**) qRT-PCR results of the expression levels of *VvWRKY28* in WT and transgenic lines (S2, S6, S7). (**B**) The phenotype of WT and transgenic plants after low temperature stress and recovery. The ruler stood for 3 cm. (**C**) Survival rates of WT and transgenic lines under normal and low temperature conditions. *t*-test showed that under cold stress, there was a very significant difference between transgenic lines and WT, ** *p* ≤ 0.01.

**Figure 5 ijms-23-13418-f005:**
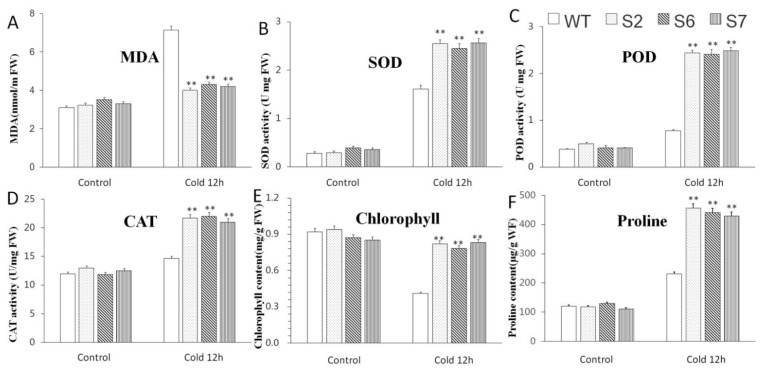
Effects of overexpression of *VvWRKY28* on physiological and biochemical indexes of *A. thaliana* under low temperature. (**A**) MDA content; (**B**) SOD activity; (**C**) POD activity; (**D**) CAT activity; (**E**) Chlorophyll content; (**F**) Proline content. The standard error was the average of 3 repeated tests. The asterisk above the column indicated that there was a very significant difference between the transgenic lines and the WT lines, ** *p* ≤ 0.01.

**Figure 6 ijms-23-13418-f006:**
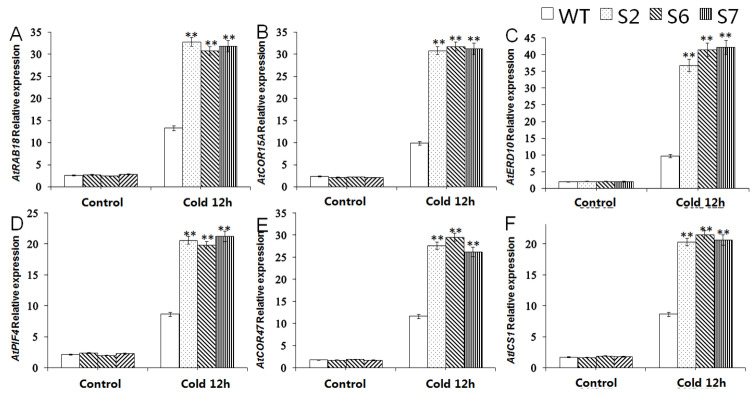
Expression levels of 6 cold stress related genes (**A**) *RAB18*, (**B**) *COR15A*, (**C**) *ERD10*, (**D**) *PIF4*, (**E**) COR47, and (**F**) *ICS1* in WT and transgenic *Arabidopsis* under low temperature stress. The standard error was the average of 3 repeated tests. The asterisk above the column indicated that there was a significant difference between the transgenic lines and the WT lines under cold stress (** *p* ≤ 0.01).

**Figure 7 ijms-23-13418-f007:**
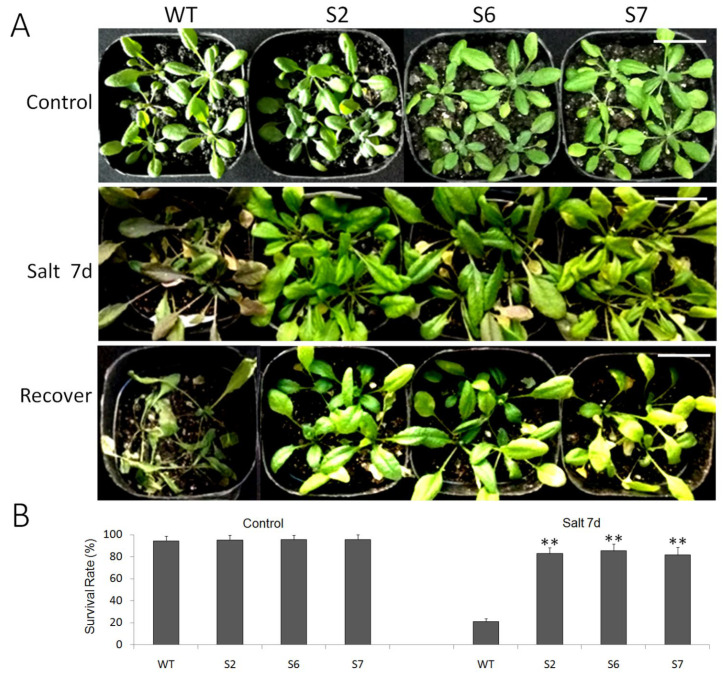
The overexpression of *VvWRKY28* improved the salt tolerance of *Arabidopsis*. (**A**) The phenotype of WT and transgenic plants after salt stress and recovery. The ruler stood for 3 cm. (**B**) Survival rates of WT and transgenic lines under normal and salt conditions. The T test showed that, under salt stress, there was a very significant difference between transgenic lines and WT, ** *p* ≤ 0.01.

**Figure 8 ijms-23-13418-f008:**
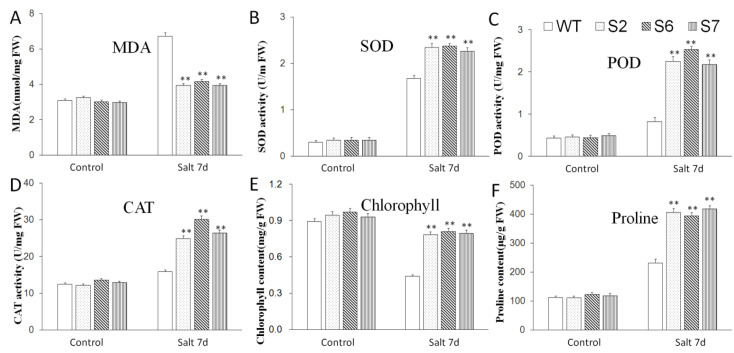
Effects of overexpression of *VvWRKY28* on physiological and biochemical indexes of *A. thaliana* under high salt conditions. (**A**) MDA content; (**B**) SOD activity; (**C**) POD activity; (**D**) CAT activity; (**E**) Chlorophyll content; (**F**) Proline content. The standard error was the average of 3 repeated tests. The asterisk above the column indicated that there was a very significant difference between the transgenic lines and the WT lines, **, *p* ≤ 0.01.

**Figure 9 ijms-23-13418-f009:**
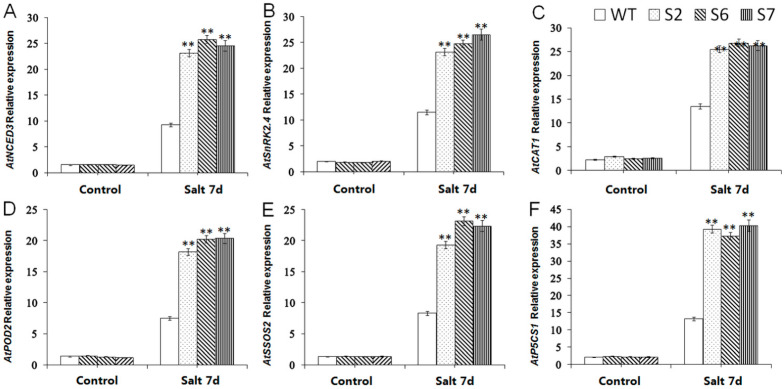
Expression levels of 6 salt stress related genes—(**A**) *NCED3*, (**B**) *SnRK2.4*, (**C**) *CAT2*, (**D**) *SOD1*, (**E**) *SOS2*, and (**F**) *P5CS1* in WT and transgenic *Arabidopsis* under salt stress. The standard error was the average of 3 repeated tests. The asterisk above the column indicated that there was a significant difference between the transgenic lines and the WT lines under salt stress (** *p* ≤ 0.01).

**Figure 10 ijms-23-13418-f010:**
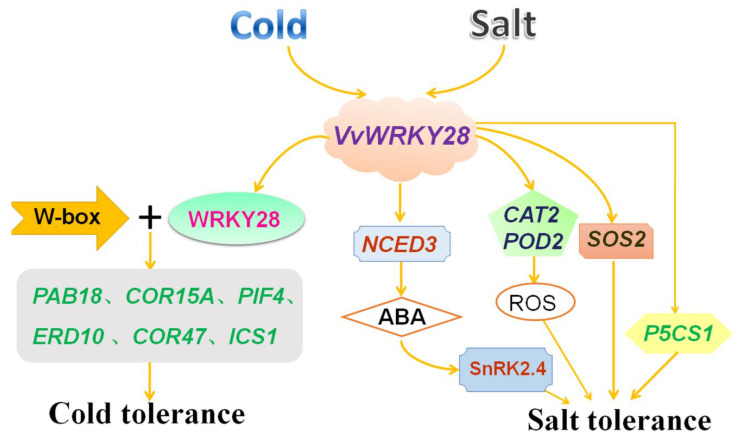
A possible model of *VvWRKY28* in response to salt stress. *VvWRKY28* was overexpressed after cold stress, binded to the W-box on the promoter of downstream target genes, and up-regulated the expression of *RAB18*, *COR15A*, *ERD10*, *PIF4*, *COR47*, and *ICS1*. Secondly, salt stress induces high-level expression of *VvWRKY28*, activates ABA signaling pathway and salt stress response process, and promotes the expression of *NCED3* and *SnRK2.4* in the ABA-dependent pathway. The expression levels of *CAT2*, *SOD1*, and *SOS2* and ABA-independent *P5CS1* genes increased significantly.

## Data Availability

The original data forthis present study are available from the corresponding authors.

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
