# Peer review of "Isolation and Functional Analysis of *VvWRKY28*, a *Vitis vinifera* WRKY Transcription Factor Gene, with Functions in Tolerance to Cold and Salt Stress in Transgenic *Arabidopsis thaliana"

_ijms, 2022, doi:10.3390/ijms232113418_

Round 1
Reviewer 1 Report
This article describes the isolations of a transcription factor putatively implicated in salt and cold stress responses. In order to validate the putative function of VvWRKY28, authors transformed A. thaliana plants. The work fits the scope of the journal. Sound conclusions and discussion are missing which difficults the assessment of the real advancement of the current knowledge.
The abstract needs to be revised in terms of language, expressions such as” for example” “obviously” are not adequate. The words “can induce high expression”, “not changed much” referring to the results are vague and inconclusive. The abstract must be a succinct description of objectives, results and conclusions.
Keywords: Some of the keywords are already included in the title, cold stress, salt stress, they need to be replaced
Introduction: This section must be reorganized, sentences are long and confusing and in some cases cut-off (According to a large number….). English syntaxis is bad, an English native speaker must revise the use of English. Acronyms must be expressed in full before using them or provide a list of abbreviations.
Results: The structure of the VvWRKY28 protein presents alignment with fourteen species, the quality of the image makes it unreadable. Besides, justification of the criterion for the selection of those species is lacking. Other V vinifera previously reported WRKY genes are not included as WRKY A. thaliana. The phylogenetic tree would have been more illustrative if the comparison was done with other TF putatively and functionally similar. No relevant conclusions are drawn from the phylogenetic tree due the apparent lack of criterion on the choice of species. That study does not add to the manuscript and is not further discussed.
There are paragraphs that are repeated in the results section (Expression Specificity Analysis of VvWRKY28).
There is no clear justification of the choice of sampling time-points. As in abiotic stresses acclimation, is an important factor, this should have been considered in the experimental design or at least addressed in the discussion. No explanation or discussion addresses the fact that the overexpression doesn’t follow a time pattern with sudden rise and decline. Experiments are planned and conducted without justification of experimental approach, they are not based on previous scientific experiments or reports.
Raw data for expression studies should be available, at least some evidence of the level of expression in regard to the reference gene used.
Discussion: There is a lack of discussion trying to decipher the real mechanisms underneath the results. The discussion is a mere summary of the results. No clear conclusions are drawn from the study besides that the TF is generally involved in salt and cold stress.
References: Number of references is too high and there are mistakes and repetitions among them.
Reviewer 2 Report
Please find the attached file to the suggested revisions.

Round 2
Reviewer 1 Report
The authors have made only slight changes to the manuscript, mainly regarding English language and organization. The discussion is still poorly developed as most of it still repeats the results section. Conclusions should be reconsidered.
